# A Concrete Core Void Imaging Approach and Parameter Analysis of Concrete-Filled Steel Tube Members Using Travel Time Tomography: Multi-Physics Simulations and Experimental Studies

**DOI:** 10.3390/s24082503

**Published:** 2024-04-13

**Authors:** Wenting Zheng, Bin Xu, Zongjun Xia, Jiang Wang, Jingliang Liu, Yudi Yao, Yifei Wang

**Affiliations:** 1College of Civil Engineering, Huaqiao University, Xiamen 361021, China; wentingzheng@stu.hqu.edu.cn; 2College of Civil Engineering, Fujian University of Technology, Fuzhou 350118, China; 3Key Laboratory for Intelligent Infrastructure and Monitoring of Fujian Province, Huaqiao University, Xiamen 361021, China; 4China Railway 15th Bureau Group City Construction Company Ltd., Luoyang 471000, China; 1120205123@tju.edu.cn (Z.X.); yaoyudi625@163.com (Y.Y.); 18303690102@163.com (Y.W.); 5Research Institute of Urbanization and Urban Safety, School of Civil and Resource Engineering, University of Science and Technology Beijing, Beijing 100083, China; 18011086007@hqu.edu.cn; 6College of Transportation and Civil Engineering, Fujian Agriculture and Forestry University, Fuzhou 350108, China; liujingliang@fafu.edu.cn

**Keywords:** concrete-filled steel tube, curved ray theory-based travel time tomography, least square iterative linear inversion algorithm, piezoelectric lead zirconate titanate, defect imaging, parameter analysis, numerical study, experimental study

## Abstract

Concrete-filled steel tube (CFST) members have been widely used in civil engineering due to their advanced mechanical properties. However, internal defects such as the concrete core voids and interface debonding in CFST structures are likely to weaken their load-carrying capacity and stiffness, which affects the safety and serviceability. Visualizing the inner defects of the concrete cores in CFST members is a critical requirement and a challenging task due to the obvious difference in the material mechanical parameters of the concrete core and steel tube in CFST members. In this study, a curved ray theory-based travel time tomography (TTT) with a least square iterative linear inversion algorithm is first introduced to quantitatively identify and visualize the sizes and positions of the concrete core voids in CFST members. Secondly, a numerical investigation of the influence of different parameters on the inversion algorithm for the defect imaging of CFST members, including the effects of the model weighting matrix, weighting factor and grid size on the void’s imaging quality and accuracy, is carried out. Finally, an experimental study on six CFST specimens with mimicked concrete core void defects is performed in a laboratory and the mimicked defects are visualized. The results demonstrate that TTT can identify the sizes and positions of the concrete core void defects in CFST members efficiently with the use of optimal parameters.

## 1. Introduction

With the development of building materials [1], construction technology and computational theory, large-scale concrete–steel composite structures have been implemented in the past few decades. Compared with traditional reinforced concrete (RC) or steel structures, concrete-filled steel tube (CFST) structures can make full use of the material strengths of concrete and steel to improve their bearing capacity and seismic resistance [2,3,4,5]. For example, circular and rectangular CFST members have often been used for super high-rise buildings, long-span bridges, harbor engineering, subway stations and other large-scale engineering structures [6,7,8]. However, CFST structures are prone to yield defects such as interface debonding and concrete core voids due to unavoidable concrete shrinkage and creep and the obvious non-uniform heat distributed due to hydration, temperature difference, construction quality issues, etc. Existing studies show that the ultimate load-carrying capacity and stiffness of CFST members are dramatically weakened due to the loss of the confinement effect of steel tubes on the concrete core and the existence of voids in the concrete core [9,10,11,12]. Therefore, it is imperative and critical to explore effective defect inspection and health condition assessment techniques for CFST members [13].

Various wave-theory-based non-destructive detection approaches for the assessment of large-scale CFST structures have been proposed and used in practice in recent years. For example, Xu et al., Chen et al. and Wang et al. [14,15,16,17,18,19,20] carried out theoretical and experimental studies to investigate the feasibility and the principles of defect detection approaches for CFST members using stress wave measurements with piezoelectric lead zirconate titanate (PZT) actuators and sensors embedded in concrete or mounted on the outer surface of the steel tubes of CFST members. Moreover, they enable the estimation of the approximate area in which the defects appear through the feature extraction of stress wave signals and previous engineering experience. In addition to wave-theory-based non-destructive detection approaches, detection methods based on vibration signals have gradually been introduced in recent years to detect the interfacial debonding of CFST members; e.g., Liu et al. [21,22] proposed a vibration-based impact acoustic detection method to qualitatively assess the degree of interfacial debonding by calculating the vibration energy ratio. However, these vibration- and wave-theory-based methods still struggle to quantitatively identify the sizes and positions of the possible internal defects in CFST members. In contrast, imaging technology is able to visually reveal the internal defects inside CFST members, which provides a good basis and reference for health monitoring, residual life evaluation, optimal design and the subsequent maintenance and reinforcement of CFST members. In a word, the imaging of the internal defects of CFST members has significant importance in both theory and practical engineering. Therefore, it is desirable to develop defect imaging technologies to visualize the defects in CFST members for quantitative defect evaluations.

With the continuous improvement in computer performance, defect imaging technology has also undergone rapid development. For example, acoustic emission (AE), as a passive technique, is able to monitor the defect propagation caused by operational or extreme loads; however, it is mostly applied to concrete structures, and its feasibility for CFST members requires extensive validation [23]. Hu et al. [24] proposed an automated three-dimensional crack detection system based on the fusion of high-precision light detection and ranging (LiDAR) and cameras for structural cracks, which possesses sub-millimeter accuracy. However, this approach is only appropriate for detecting the surface cracks of structural members. In contrast, the tomography imaging method is able to gain insights to internal defects using ultrasonic technology and, therefore, it has attracted more and more attention in the nondestructive testing of building structures [25]. Typically, tomography techniques, including attenuation tomography (AT) [26] and travel time tomography (TTT) [27,28], are able to detect defects in concrete structures composed of a single material. However, strong scattering attenuation may be trigged by the concrete’s inhomogeneity at the scale of ultrasonic wavelengths, which will hinder the application of AT in large-scale engineering structures such as the CFST members in high-rise buildings or long-span bridges. In contrast, TTT has been widely applied for subsurface medium detection. For instance, a cross-hole tomography method was proposed, based on the straight ray tracing algorithm, and works for structures with homogeneous materials [29]. When the medium in a structure is not homogeneous, an obvious imaging error occurs if a straight ray tracing algorithm is employed. To overcome this shortcoming, Clement et al. [30] proposed a cross-hole radar TTT algorithm based on a curved ray tracing method. This method is essentially an iterative scheme based on two-dimensional (2D) curved ray tracing, which can successfully provide a more accurate 2D image of the subsurface structure than the method based on straight ray tracing technology. Therefore, the curved ray tracing-based TTT method has been widely accepted and gradually extended from geophysical exploration to the non-destructive inspection of civil engineering structures [31,32,33,34,35,36]. However, current TTT has been used for concrete structures mostly [37,38] and limited studies have been carried out on the defect visualization of CFST members composed of two different kinds of construction materials with different mechanical properties [39,40,41]. Actually, the travel time of stress waves depends on the elastic modulus and density of their propagation medium. Compared to traditional concrete structures with a single concrete material, the propagation process of stress waves through CFST members is more complicated, which increases the difficulty in imaging the internal defects of CFST members. Therefore, how to accurately track the travel paths of stress waves inside steel tubes and concrete cores and acquire their travel velocity distribution becomes a critical issue for the quantification of the internal defects inside CFST members. 

In order to address these issues mentioned above, stress wave response signals and their travel times are obtained through wave measurements, and then the curved ray theory-based TTT algorithm is applied to these signals to visualize the concrete core void defects of rectangular CFST members. Moreover, a parameter analysis is performed to investigate the effect of the parameters in the reverse algorithm on the quality of the imaging of CFST members with different sizes and locations of void defects. Then, several optimal parameters are recommended for the visualization of the inner concrete core voids in rectangular CFST members using the proposed reverse approach. Numerical simulation results demonstrate that the proposed approach is capable of identifying the position and size of void defects in concrete cores effectively. Finally, an experimental study on six CFST specimens with various concrete core void defects is carried out, and the imaging results using the measured stress wave responses are compared with the real concrete core voids. As such, the effectiveness and accuracy of the curved ray theory-based TTT method for the concrete core void imaging of CFST members are demonstrated.

The outline of this paper is as follows: The curved ray theory-based TTT algorithm is detailed in Section 2, while the multi-physics coupling finite element model (FEM) of PZT-CFST is established in Section 3. The analysis of the parameters of the reverse algorithm for the imaging and the generalization of the proposed imaging approach for rectangular CFST members with different concrete core void sizes and positions are investigated numerically in Section 4 and Section 5, respectively. Experimental verification is carried out for CFST specimens with mimicked concrete core void defects in Section 6. Concluding remarks are given in Section 7.

## 2. Curved Ray Theory-Based TTT for the Concrete Core Void Imaging of CFST Members Using the Least Square Iterative Linear Inversion Algorithm

In this section, curved ray theory-based TTT is proposed to obtain the optimum solution for the velocity/slowness distributions within the cross section of a CFST member between actuators and sensors mounted on the outer surface of the CFST member, according to the first arrival times of the stress waves. A flowchart of the curved ray theory-based TTT method is given in Figure 1. As shown in Figure 1, a ray-tracing technique is required in TTT. Typically, when the propagation medium of the stress wave is assumed to be homogeneous, the ray path between an arbitrary actuator and sensor can be approximately assumed to be a straight ray. Figure 2 shows the schematic diagram of the ray path tracing for the cross section of a CFST member. As shown in Figure 2a, *n* actuators and *n* sensors are mounted on the outer surface of the two opposite sides of the cross section of the CFST member being visualized. Tx and Rx represent the labels/names of the actuators and sensors, respectively. If an excitation signal is generated by an arbitrary actuator, each of *n* sensors on the opposite side will receive a distinct stress wave that has travelled through the cross section of the CFST member and, as a result, *n* straight rays are yielded. Similarly, *n* × *n* straight rays will be received if all *n* actuators are excited. Nevertheless, the traditional straight ray theorem is no longer suitable for simulating the transmission of stress waves inside CFST members with void defects in their concrete core due to the different structural mechanical properties of their steel, concrete core and voids. To be specific, stress waves are likely to bypass the internal concrete core void defects/abnormal low-velocity mediums and turn to travel along the path that takes the least time. In contrast, the curved ray tracing theory provides more realistic ray tracing paths for CFST members than the typical straight ray method and yields a more accurate velocity distribution as well.

According to the principle of TTT, an arbitrary curved ray path can be regarded as a sum of several straight rays when that path is divided by the discrete slowness grids of the cross section of a CFST member, as illustrated in Figure 2b. Actually, the discrete slowness grids are square areas that are yielded by meshing the cross section of a CFST member. Here, slowness indicates the reciprocal of velocity. That is to say, the size of a slowness grid, e.g., S1, equals 1/2*^p^* (*p* = 0, 1, 2, 3, …) times the distance between adjacent actuators/sensors. As a result, the travel time of a stress wave along the curved ray path can be obtained by summing up the parts of all straight rays that constitute it. For example, the time spent travelling along the *i*-th curved ray, expressed as $t_{i}$, is calculated according to Equation (1).
(1)$$t_{i}=\sum_{j=1}^{m}{l_{j}^{i}s}_{j}$$
where *i* and *j* denote the *i*-th curved ray and the *j*-th grid, respectively. *m* represents the total number of grids that the *i*-th ray passes through. Here, $s_{j}$ is the slowness inside the *j*-th grid, while $l_{j}^{i}$ represents the length of the *i*-th curved ray within the *j*-th grid.

If *n* actuators and *n* sensors are arranged on the two opposite sides (left and right sides in Figure 2) of a CFST member, the total number of rays will equal *n* × *n*. Then, the calculated travel time matrix, as expressed in Equation (2), can be established on the basis of Equation (1).
(2)$$t_{cal}=t_{n\times n}=L_{n\times m}s_{m\times n}$$
where $t_{cal}$ and s represent the matrices of the calculated travel time and slowness, respectively. Here, L is a coefficient matrix containing the lengths of the curved ray paths that pass through the slowness grids. So, each element in L can be represented by $l_{j}^{i}$.

Due to the fact that ***L*** is usually a sparse matrix, it has the possibility to be a singular matrix, which implies that the inverse matrix of $L^{-1}$ does not exist. In order to address this issue, a Tikhonov regularization function [34,42] is introduced to define the objective function, as in Equation (3).
(3)$$\Phi \left(s\right)={\left\|W_{d}\left[t-Ls\right]\right\|}^{2}+\lambda^{2}{\left\|W_{m}\left[s-s^{0}\right]\right\|}^{2}$$
where λ is weighting factor. $W_{d}$ and $W_{m}$ represent the data and model weighting matrices, respectively. s and $s^{0}$, separately, represent the slowness matrix to be solved and the initial slowness matrix. Here, $s^{0}$ is often defined as the slowness distribution of the cross section of the CFST member under its intact state. Each element in $s^{0}$ is solved by $s^{0}=\sqrt{\rho /E}$, with ρ and E representing the density and elastic modulus of steel and concrete, respectively. As symmetric matrices, the a priori information contained in $W_{d}$ and $W_{m}$ can effectively guarantee the uniqueness of the solution to the nonlinear equation shown in Equation (3).

It can be seen from Equation (3) that the coefficient matrix ***L*** is actually a function of the slowness matrix s on condition that ***t*** is solved by employing the first arrival time of the stress wave, as mentioned before. Therefore, Ls can be expressed as d(s) and then is defined as the calculated travel time matrix as well. After that, it can be solved using the finite difference method. To minimize Equation (3), the partial derivative of s with respect to time is calculated, and hence Equation (3) can be expressed as Equation (4) [34].
(4)$$\lambda^{2}W_{m}^{T}W_{m}\left(s-s^{0}\right)=J^{T}W_{d}^{T}W_{d}\left[t-d(s)\right]$$
where J=∂d(s)/∂s denotes the Jacobian matrix, while the subscript of *T* represents the transpose operation of a matrix.

With an understanding of the nonlinear properties of d(s), ***J*** cannot be obtained directly and the initial Jacobian matrix $J^{0}$ can only be solved by knowing the initial slowness matrix $s^{0}$ in advance. Hence, an iterative algorithm is required to solve Equation (4). Typically, a linear iteration method is used to create a Taylor series expansion of d(s), and the first-order term is retained alone, leading to $d\left(s\right)$ being defined as illustrated in Equation (5).
(5)$$d\left(s\right)\approx d\left(s^{0}\right)+J^{0}\left(s-s^{0}\right)$$

As such, Equation (4) becomes
(6)$$\left[{\left(J^{0}\right)}^{T}W_{d}^{T}W_{d}J^{0}+\lambda^{2}W_{m}^{T}W_{m}\right]\left(s-s^{0}\right)={\left(J^{0}\right)}^{T}W_{d}^{T}W_{d}\left[t-d(s^{0})\right]$$

It should be noted here that J is replaced by $J^{0}$ in Equation (6) due to it only denoting the initial iteration step. After that, Equation (6) can be rewritten in its iterative form and presented as Equation (7) [34].
(7)$$\left[{\left(J^{k}\right)}^{T}W_{d}^{T}W_{d}J^{k}+\lambda^{2}W_{m}^{T}W_{m}\right]\left(s^{k+1}-s^{k}\right)={\left(J^{k}\right)}^{T}W_{d}^{T}W_{d}\left[t-d\left(s^{k}\right)\right],\;\left(k=0,1,2,3\cdots \right)$$
where *k* represents the number of iterations.

The updated slowness matrix $s^{k}$ can be solved according to Equation (7). Normally, it is iterated repeatedly until the error between the calculated travel time matrix $d(s^{k})$ and the measured travel time matrix ***t*** reaches a determined threshold or the number of iterations exceeds a number defined beforehand. Thus, the final updated slowness matrix $s^{k+1}$ can be regarded as the actual slowness field inside the CFST member.

To enhance the efficiency of solving Equation (7), the least square algorithm is used instead of applying the above-mentioned iterative approach alone. Therefore, Equation (7) is equivalent to Equation (8), and the process of solving Equation (8) by the least square method is called the least square iterative linear inversion algorithm [43].
(8)WdJkWmsk+1−sk=Wdt−dsk0,(k=0,1,2,3⋯)
where $W_{d}$, $W_{m}$ and λ represent the data weighting matrix, model weighting matrix and weighting factor, respectively.
(9)$$J=\frac{\partial d(s)}{\partial s}=\frac{\partial Ls}{\partial s}=L$$

As seen in Equation (9), the Jacobian matrix ***J*** equals the coefficient matrix ***L*** according to the curved ray theory-based TTT method. Therefore, the Jacobian matrix ***J*** can be established by calculating $l_{j}^{i}$.

## 3. Multi-Physics FEM for the Stress Wave Travel Time Determination of a PZT-CFST Coupling System

For the successful application of curved ray theory-based TTT, the travel time of the stress waves from each actuator to each sensor should be determined accurately. A multi-physics coupling FEM, composed of a PZT-CFST with a concrete core void, is constructed and illustrated in Figure 3. The sizes of the rectangular CFST model and the concrete core void defect are 400 mm × 400 mm and 120 mm × 80 mm, respectively, while the thickness of the surrounding steel tube is 5 mm. A total of 21 PZT patches are arranged as actuators, with an equal interval of 20 mm (0.02 m), on the left side of the outer surface of the steel tube, while another 21 PZT patches, used as sensors, are laid out on the outer surface of the steel tube on the opposite side. The actuators are labeled T1 to T21 from the left upper to left lower side, while the sensors are named R1 to R21 in a similar way, from the right upper to right lower side. The planar size of each PZT patch is 10 mm × 10 mm and their thickness is 0.3 mm. Here, the polarization direction of the employed PZT patch is along the thickness direction. The material properties of the PZT-CFST coupling model are defined according to [19,44,45] and illustrated in Table 1.

At first, a sinusoidal signal with a frequency of 20 kHz and an amplitude of 10 V is set as the excitation signal applied to the PZT actuators and the signal’s waveform is plotted in Figure 4a. Here, four cycles of the excitation signal are selected for signal processing; that is to say, the excitation signal has at least a time duration of 4 × 1/20 = 0.2 ms. The stress wave propagates in the CFST section with a concrete core void defect and the resultant response signal is received by a sensor. Taking the stress wave with the path of T6-R21 as an example, the comparison of its response signals in an intact CFST member and a CFST member with concrete core void defect is shown in Figure 4b. In order to extract the first wave peak time from the measured response signal more effectively, a peak picking method similar to the extraction of modal frequency peaks from Fourier spectra is used [46]. As seen in Figure 4b, the red dashed line indicates the received response signal after the stress wave travels through the cross section of the CFST model with a concrete core void defect, and the red pentagram represents the first wave peak arrival time of the received stress wave response signal from the CFST model. Compared with the first wave peak arrival time of the intact CFST model (black diamond), its counterpart (red pentagram) in the CFST model with a concrete core void defect is slightly delayed and its amplitude decreases to an extent as well. The reason for this phenomenon is that the travel path changes when the stress wave encounters a concrete core void defect. In other words, the stress wave is likely to propagate along the edge of the concrete core void instead of traveling through the concrete core void itself. As a result, the travel path becomes relatively longer, leading to the wave’s delayed travel time and energy attenuation as well.

Following the extraction of the first wave peak arrival time, the time interval between the first wave peak arrival time of the PZT sensor measurement signal and that of the excitation signal is defined as the travel time of the stress wave. Since 21 actuators and 21 sensors are separately arranged on the left and right sides of the cross section, 441 travel paths (21 × 21 = 441) will be yielded and, thus, the same number of first wave travel times will be extracted accordingly. After that, the 441 extracted first wave travel time data points constitute a measured travel time matrix (t) and this is then substituted into Equation (8) for iteration and an inversion analysis.

## 4. Analysis of the Effects of the Parameters of the Inversion Analysis on the Defect Imaging of CFST Members

In curved ray theory-based TTT, the employed default inversion algorithm for imaging is the least square QR (LSQR). Here, QR indicates QR decomposition.

According to Equation (8), the parameters of $W_{d}$, $W_{m}$ and λ in the inversion algorithm are artificially defined and hence the imaging results are possibly affected by $W_{d}$, $W_{m}$ and λ. That is to say, they are the main three factors used to balance the travel time residual of $\left[t-Ls\right]$ and the slowness residual of $\left[s-s^{0}\right]$ in the objective function of $\Phi \left(s\right)$. In addition, the grid size (*d*) not only affects the computational efficiency of curved ray theory-based TTT, but also influences the accuracy of the imaging results to an extent. Dong et al. and Liu et al. have carried out preliminary attempts to verify the effectiveness of TTT on CFST members [39,40]; however, a detailed investigation of how the parameters of the inversion algorithm for the defect imaging of CFST members affect the imaging quality and accuracy of CFST members with different void defect numbers, sizes and positions is required.

### 4.1. Selection of Parameters Used in Reverse Algorithms

As mentioned before, $W_{d}$, $W_{m}$ and λ are the three parameters affecting the defect imaging results. Normally, the unit matrix operator is used for $W_{d}$, which indicates that the data are uniformly weighted. Due to this, $W_{d}$ does not affect the imaging results too extensively and hence its effect is no longer considered here. In contrast, the influence of other parameters, including the model weighting matrix ($W_{m}$) and weighting factor (λ), is investigated in addition to the grid size (*d*).

In order to better understand the influence of $W_{m}$, λ and *d* on the TTT results of the concrete core void defects inside the CFST cross section, as illustrated in Figure 3, a total of seven cases are considered, as shown in Table 2. To quantify the imaging effects of void defects under different parameters, the root mean square error (RMSE) between the measured travel time ***t*** and calculated travel time ***d***(***s***) is computed via Equation (10). The smaller the RMSE, the more accurate the calculated travel time.
(10)$$RMSE=\sqrt{\frac{1}{n}\sum_{1}^{n}{\left|t-d(s)\right|}^{2}}$$
where *n* represents the number of actuators or sensors.

### 4.2. Model Weighting Matrix

Since the model weighting matrix $W_{m}$ contains prior information to reduce the multiple solutions to inversion problems, its selection is vital for concrete core void defect imaging. By selecting different $W_{m}$, various constraints are applied to the residual term $\left[s-s^{0}\right]$ to control the precision of the imaging results generated by TTT. Three types of model weighting matrices, including the unit matrix operator, first-order difference operator and second-order difference operator, are considered.

Through the use of curved ray theory-based TTT via the least square iterative linear inversion algorithm, the imaging results of the concrete core void defect of the rectangular CFST model are illustrated in Figure 5 with the use of different model weighting matrices. It is noted here that all void defect imaging results in the rest of the sections of this paper are plotted in the style of a greyscale map. That is to say, the white color in the greyscale maps represents the concrete core void defect, while the grey and black colors denote concrete cores and steel tubes, respectively. X and Y in the greyscale map, respectively, represent the horizontal and vertical coordinates of the cross section of CFST members with void defects. In addition, the numbers on the color bar denote the identified velocities using the proposed approach (unit: m/s). 

As shown in Figure 5, the outer steel tubes are imaged effectively in all cases when the initial slowness matrix is defined according to the slowness distribution of the intact CFST model. It can be seen from Figure 5a that the position of the concrete core void defect basically agrees with the theoretical result when the unit matrix operator is selected; however, the size of the concrete core void defect is not accurately imaged in Case 1. Moreover, a few white spots appear around the concrete void and steel tube, and most of the white spots are likely to be mistakes. When the first-order difference operator is employed (Case 2), the size of the concrete core void defect’s image is larger than that of the actual model. In addition to this, the imaged shape of the concrete core void defect is also slightly distorted in Case 2. It can be seen from Figure 5c that the position and size of the concrete core void defect agree well with the theoretical result in Case 3, when compared to Figure 5a,b. Also, Figure 6 presents an illustration of the RMSE with different $W_{m}$. It can be seen from Figure 6 that the RMSE in Case 3 is the smallest when compared with its counterparts in Case 1 and Case 2. That is to say, the calculated travel time in Case 3 is the most accurate, which will result in a more accurate void imaging result via TTT. In a word, TTT has the potential to image rectangular CFST members with acceptable accuracy when the second-order difference operator is selected for modeling.

### 4.3. Weighting Factor

The weighting factor (λ) controls the weight between the data fitting term (${\left\|W_{d}\left[t-Ls\right]\right\|}^{2}$) and the regularization term (${\left\|W_{m}\left[s-s^{0}\right]\right\|}^{2}$), leading to different regularization effects. To be specific, a large λ probably leads to a heavier restriction of the model parameters, making the model simpler and reducing the risk of overfitting. However, it might ignore some detailed information, resulting in smoother imaging results with a lower resolution. In contrast, a small λ would result in less restriction of the model parameters, preserving more detailed information. However, this might increase the risk of overfitting and cause noisy or unstable imaging results. Therefore, the optimum λ needs to be determined for the reverse TTT algorithm. As suggested in [34], the ***L*** curve method is recommended for determining the optimum value range of λ, which is usually between 2 and 20. For simplicity, λ = 2 (Case 4), λ = 5 (Case 3) and λ = 10 (Case 5) are investigated in this study.

The imaging results for Cases 3, 4 and 5 are plotted in Figure 7. It can be seen from Figure 7 that, in Case 4 (λ = 2), the concrete core void defect imaging result for the CFST section does not agree well with the real concrete core void defect. Moreover, two fake images of concrete core voids appear on the upper and lower sides of the central concrete core void defect in the CFST section. In contrast, the imaging results of the concrete core void defect are consistent with the real size and position of the concrete core void defect when λ = 5 (Case 3). Compared with Case 3, the imaging effect in Case 5 (λ = 10) is worse, since the size of the imaged concrete core void defect is smaller than its actual size. The reason for this phenomenon is that the model is smoothed excessively due to its large weighting factor (λ = 10), which leads to an obvious reduction in fake images. In contrast, fake images will be acquired if a small λ is selected. The void defect imaging result obtained with the proposed approach in Case 3 agrees best with the true velocity field distribution, and hence λ = 5 can be thought of as the optimum value for the concrete core void imaging of the numerical example.

Similar to Section 4.2, the index of the RMSE with the various λs mentioned above is calculated according to Equation (10), and the corresponding results are shown in Figure 8. As indicated in Figure 8, the RMSE of Case 3 (λ = 5) has the smallest value, which implies that the best void imaging result is achieved by TTT in Case 3.

### 4.4. Grid Size

Considering the fact that the thickness of the steel tube in these numerical models is constant, the grid size (*d*) of the concrete core is investigated alone in this section. During the process of TTT, the selection of the grid size (*d*) for the concrete core is related to the desired resolution for the imaging of concrete core voids. Since an arbitrary curved ray path consists of several straight rays, according to the principle of TTT, the tracing accuracy of the final travel paths greatly depend on the grid size (*d*). For example, if the grid size is large, the number of discrete grids in the concrete core of a CFST member decreases greatly, leading to an imprecise travel time calculation and poor ultimate imaging quality. On the other hand, a smaller grid size means a significant increase in computing time. Moreover, an excessively small grid size may also make the inversion problem unstable and thereby compromise the reliability and accuracy of the inversion.

In order to investigate the influence of the grid size of the concrete core on the imaging results, three grid sizes (*d*) are considered: 2 cm (Case 6), 1cm (Case 3) and 0.5 cm (Case 7). Figure 9 shows the imaging results for concrete core void defects of a CFST model using various grid sizes. As shown in Figure 9b, the position of the concrete core void defect within the CFST section is effectively imaged when *d* is 2 cm (Case 6); however, the size of the concrete core void defect is a little larger than its predefined size. In addition, a few fake defect images outside of the real void defect are generated in Case 6. This suggests that a large grid size will reduce imaging accuracy and increase the likelihood of generating false abnormal velocity distribution areas. When the grid size is 1 cm (Case 3), the imaging quality of the CFST section becomes better. The position and size of the concrete core void defect align closely with the real model in Figure 9a (Case 3). Figure 9c also illustrates that a few fake images appear in Case 7 (*d* = 0.5 cm) due to the interference of more rays with large angles. In addition to a worse concrete core void imaging result than its counterpart in Figure 9a, computing efficiency is also reduced in Case 7. Therefore, both excessively large and small grid sizes for the concrete core can lead to poor imaging quality, and hence an appropriate grid size is required. Based on the imaging results illustrated in Figure 9, a grid size of *d* = 1 cm is recommended for this numerical example.

To quantify the imaging effects on void defects of the above-mentioned three different grid sizes, the RMSE between the measured travel time ***t*** and the calculated travel time ***d***(***s***) is computed via Equation (10). The results in Figure 10 show that the RMSE is the smallest when *d* = 1 cm, and hence this is when the best void imaging result is generated by TTT.

In summary, it can be seen from Figure 6, Figure 8 and Figure 10 that the RMSE of Case 3 is the smallest, when compared with that of other cases. That is to say, the calculated travel time in Case 3 has the best accuracy, which probably leads to the best imaging results for CFST members.

## 5. Generalizability of the Proposed Imaging Approach for the Concrete Core Voids of CFST Members, Considering Void Size and Position

In this section, the generalizability of the proposed imaging approach for CFST members with different numbers of void defects that are of different sizes and at different locations is discussed.

### 5.1. Imaging Results Considering Different Concrete Core Void Sizes

In order to investigate the imaging results of CFST members with different sizes of concrete core void defects using the proposed approach, three sizes of concrete core void defects at the center of a multi-physics FEM of the PZT-CFST coupling system are considered. The dimensions of the considered concrete core void defects are 120 mm × 80 mm, 100 mm × 60 mm and 80 mm × 40 mm, respectively, which correspond to concrete core void ratios of 6%, 3.75% and 2%, respectively.

Based on the results of the parameter analysis in Section 4, the second-order difference operator, 5 and 1 cm are individually selected for the $W_{m}$, λ and the grid size of concrete core. The imaging results for all three CFST models with different void sizes are presented in Figure 11.

It can be seen from Figure 11 that the positions and sizes of the three concrete core void defects are in good accordance with those of the designed void defects in the numerical models, as shown by the red dashed lines. The results show that the proposed approach, using optimal parameters, can visualize concrete core void defects of different sizes with acceptable accuracy.

### 5.2. Imaging Results Considering Different Concrete Core Void Defect Locations

To further investigate the feasibility of the proposed imaging approach for the detection concrete core void defects at various locations within CFST members, imaging results for three CFST members with concrete core void defects at different locations have been taken into account. Each concrete core void defect measures 100 mm × 60 mm (concrete core void ratio = 3.75%). Three distinct defect locations are considered, that is, the center, the left upper/actuator side and the right lower/sensor side. The imaging results are presented in Figure 12.

As shown in Figure 12, the imaging of the concrete core void defect at the center of the cross section is in a good accordance with its actual size and location. However, the imaging quality becomes slightly worse when the only void is located in the left upper or right lower regions of the CFST member. The reason for this is that a smaller number of curve rays pass by the defects when they are close to the corner of the cross section of the CFST members. Although the imaging results for the two single voids close to the corner are not perfect, each concrete core void defect in the CFST member can be imaged with acceptable accuracy by the proposed approach.

### 5.3. Imaging Results Considering Different Numbers of Concrete Core Void Defects

Given the fact that multiple concrete core void defects usually appear in actual CFST members, a further investigation of the imaging results for CFST members with more than one concrete core defect is carried out. In this instance, we consider two concrete core void defects with an identical size of 100 mm × 60 mm but located in different positions in the cross section of a CFST member. Three distinct void configurations are considered, as indicated by the red dashed lines in Figure 13. The imaging results for these void defects, specifically those located near the left/actuator side, near the right/sensor side and along a diagonal line, are shown in Figure 13.

It can be seen from Figure 13 that the positions of the two concrete core void defects agree well with the exact positions designated in the numerical models, whether the two concrete core void defects are close to the actuator side or the sensor side. Moreover, the proposed approach is also capable of imaging the sizes of the two concrete core void defects effectively when they are located in a diagonal line on the cross section.

## 6. Experimental Verification Using CFST Specimens with Mimicked Concrete Core Void Defects

In this section, an experimental verification of the feasibility of the proposed approach for the imaging of CFST specimens with different numbers, sizes and locations of concrete core voids is carried out. The test setup is shown in Figure 14. As shown in Figure 14, a number of PZT patches are mounted on the outer surface of two opposite sides of the steel tube of the CFST specimens. The PZT patches on the upper side are used as actuators, to produce stress waves, and the PZT patches on the lower side are used as sensors to measure the stress wave’s travel time from a PZT actuator to a PZT sensor located on the opposite side of the CFST specimen. A signal generator and voltage amplifier are used to generate and amplify excitation signals and a digital oscilloscope is used to record the responses via the sensors. The PZT patches are compressive-type with polarization in their thickness direction.

Six CFST specimens, as illustrated in Figure 15, labelled CFST-S1, CFST-S2, CFST-S3, CFST-S4, CFST-S5 and CFST-S6, were tested. The Q345 steel material is used as the steel tube, and C30 concrete is employed as the concrete core of each specimen. All six specimens have identical planar dimensions of 400 mm × 400 mm, while the dimensions of their concrete core are 390 mm by 390 mm. The thickness of the steel tube is 5mm and the height of the specimens is 15mm. However, the number, size and location of the inner void defects in each specimen are totally different. As shown in Figure 15a–c, the dimensions of the void defects in CFST-S1, CFST-S2 and CFST-S3 are 120 mm × 80 mm × 15 mm, 100 mm × 60 mm × 15 mm and 80 mm × 40 mm × 15 mm, respectively. All of the voids are located at the center of the three CFST specimens. The void defects in CFST-S4, CFST-S5 and CFST-S6 have identical dimensions of 100 mm × 60 mm × 15 mm. However, the location and number of these voids are different.

For each CFST specimen, 21 PZT patches are mounted on two opposite sides of the specimen, with equal intervals. Insulation between the PZT patches and the steel tubes should be maintained. The size of the PZT patches is 10 mm × 10 mm × 0.3 mm, and their positive and negative electrodes are on the same surface. One PZT patch on one side of the specimen is used as an actuator and the response of all twenty-one PZT sensors on the other side are recorded to determine the wave’s travel time from the PZT actuator to each PZT sensor. The excitation, as shown in Figure 16a, is a sinusoidal impulse modulation signal with a natural frequency of 10 kHz (its period is 100 μs) and an amplitude of 11.5 V. The sampling frequency of the measured stress wave response signals is 50,000 kHz. After that, a Butterworth filter is introduced to process the measured response signals. For simplicity, only one of the processed response signals from CFST-S3 is illustrated in Figure 16b. As shown in Figure 16b, the first wave peak arrival time (represented by the red pentagram) can be easily extracted using the peak picking method mentioned in Section 3, and the corresponding stress wave travel time from each PZT actuator to each PZT sensor can be determined. Therefore, the imaging of the void defects in each specimen is achieved using the proposed approach.

The imaging results of the six CFST specimens are shown in Figure 17. Figure 17 demonstrates that the outer steel tubes of all CFST specimens are accurately imaged. As illustrated in Figure 17a–c, the single void defect in these three specimens can be imaged with acceptable accuracy using the proposed approach and the measured stress wave travel data from the test. The size of each concrete core void is slightly bigger than their actual counterparts. The imaging results for the void defects of specimens CFST-S4, CFST-S5 and CFST-S6, compared with the red dashed line showing the actual defects’ locations, are illustrated in Figure 17d–f. It is clear that the proposed imaging algorithm can visualize the sizes and locations of concrete core void defects effectively with acceptable accuracy no matter where the defects are located.

## 7. Concluding Remarks

The curved ray theory-based TTT using a least square iterative linear inversion algorithm is proposed to image the sizes and locations of concrete core void defects in CFST members using surface-mounted PZT actuating and sensing technology. In addition, a parameter analysis is also performed to reflect how the parameters of the inversion analysis and the defects’ sizes and positions affect the imaging results. In order to verify the effectiveness and accuracy of the proposed algorithm, several numerical examples and six CFST specimens with different numbers of void defects at different locations are investigated. Based on our numerical and experimental study, the following conclusions can be drawn:

(1) The numerical simulation results using multi-physics FEM models of the coupling system composed of PZT patches and CFST members demonstrate that the proposed approach can effectively visualize the sizes and locations of the concrete core void defects in each rectangular CFST model, its steel tube and its concrete core with acceptable accuracy.

(2) The influence of the model weighting matrix ($W_{m}$), weighting factor (λ) and the grid size (*d*) on the imaging results is investigated in detail. The results show that the imaging quality is best if the second-order difference operator is selected as the model weighting matrix. The optimal values for the weighting factor (λ) and grid size (*d*) are also suggested.

(3) The imaging results of six CFST specimens with different numbers, sizes and locations of voids are described in detail using the stress wave travel time measurements from our tests. The experimental results show that the number, size and location of each void defect in each CFST specimen can be visualized with acceptable accuracy.

It should be noted here that preliminary numerical and experimental study results on the void imaging of rectangular CFST members are described in this paper. There are still some limitations, as follows, remaining in the proposed approach: (1) the imaging quality heavily depends on the coverage rate of the curved rays that travel across the cross section of the CFST members; (2) the accuracy of the proposed method heavily depends on the extraction of the travel time from the stress wave response signals. In view of the vibration-based inspection technology that has been widely applied in the void defect detection of CFST members, its combination with this TTT method is likely to be a promising approach in the near future. In addition to this, further investigations on the feasibility of the proposed approach for two-dimensional circular CFST members and three-dimensional rectangular or circular CFST members are desirable. Moreover, extensive study on the feasibility of this approach for the defect visualization of CFST members whose concrete core is modelled as a mesoscale heterogeneous material should be carried out. 

## Figures and Tables

**Figure 1 sensors-24-02503-f001:**
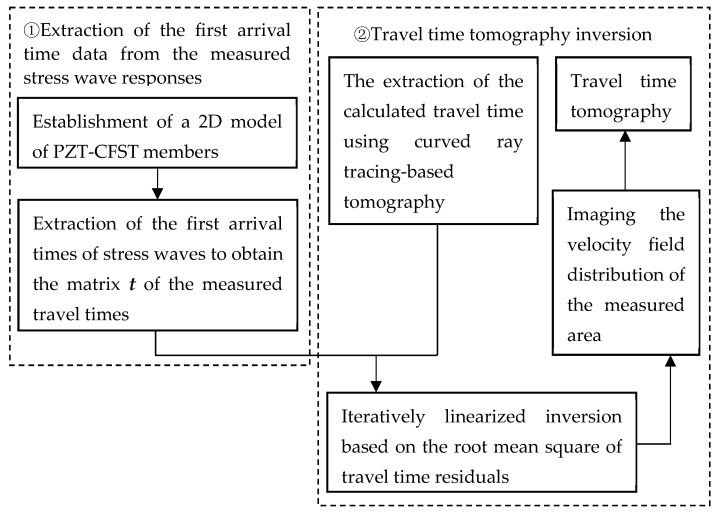
The flowchart of the curved ray theory-based TTT.

**Figure 2 sensors-24-02503-f002:**
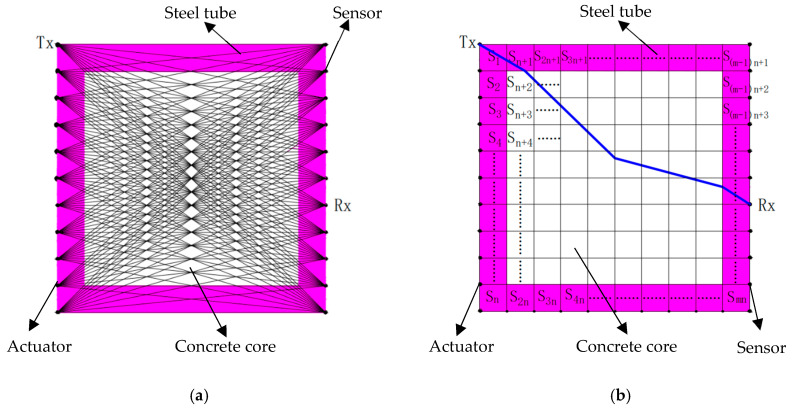
The schematic diagrams of ray path tracing for the cross section of a CFST member. (**a**) all straight ray paths; (**b**) a curved ray path divided into discrete slowness grids.

**Figure 3 sensors-24-02503-f003:**
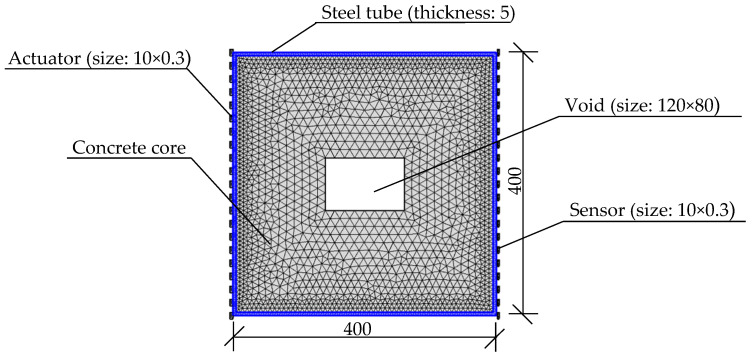
Multi-physics coupling FEM of PZT-CFST system (unit: mm).

**Figure 4 sensors-24-02503-f004:**
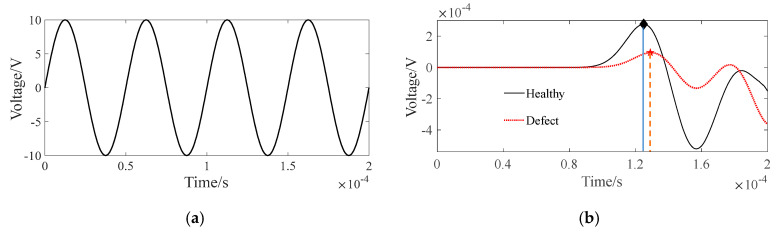
Sinusoidal excitation and stress wave response signal. (**a**) Sinusoidal excitation; (**b**) stress wave response of T6-R21.

**Figure 5 sensors-24-02503-f005:**
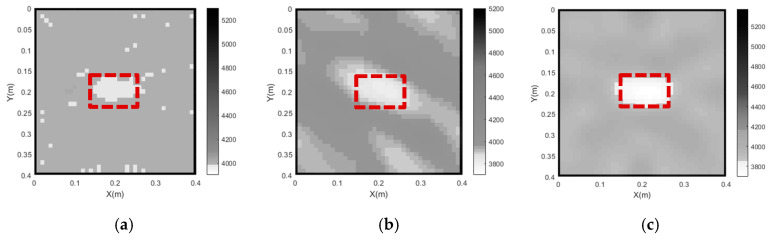
Imaging results of concrete core void defects of a CFST model under various model weighting matrices. (**a**) Case 1; (**b**) Case 2; (**c**) Case 3.

**Figure 6 sensors-24-02503-f006:**
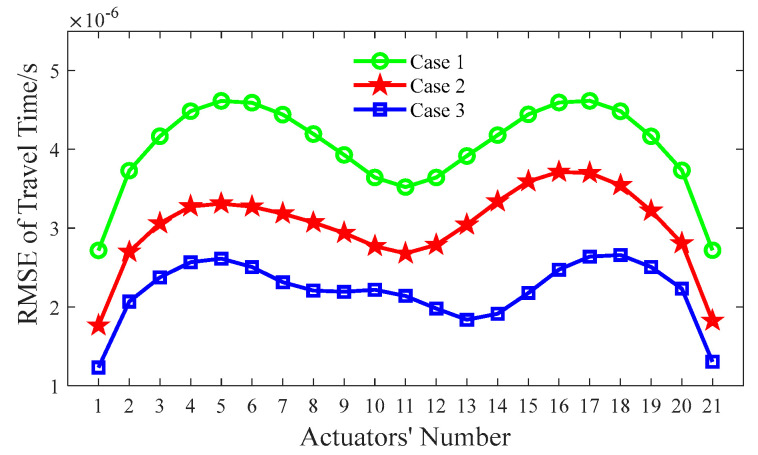
RMSE of travel time with various $W_{m}$.

**Figure 7 sensors-24-02503-f007:**
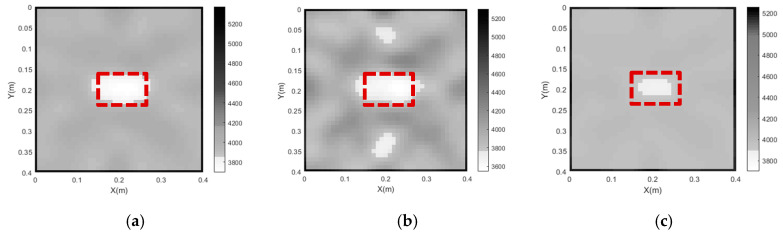
Imaging results of concrete core void defects of a CFST model using various weighting factors. (**a**) Case 3 (λ = 5); (**b**) Case 4 (λ = 2); (**c**) Case 5 (λ = 10).

**Figure 8 sensors-24-02503-f008:**
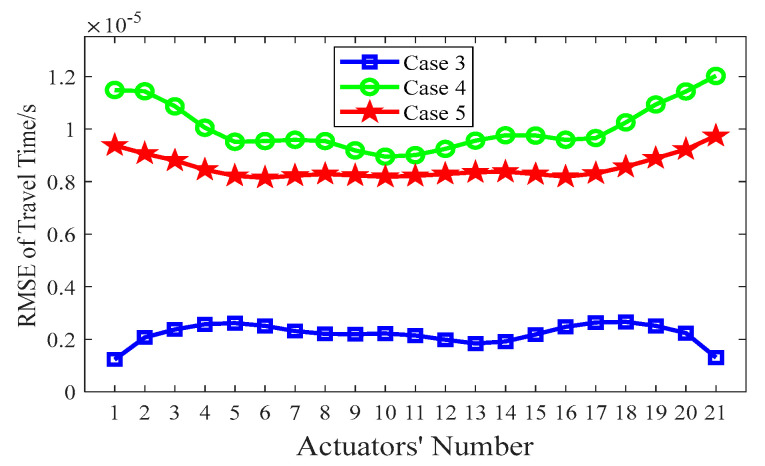
RMSE of travel time with various λs.

**Figure 9 sensors-24-02503-f009:**
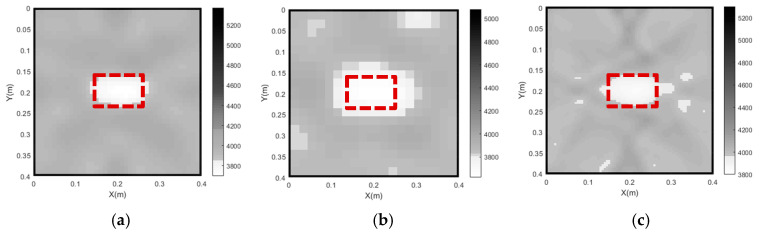
Imaging results of the concrete core void defects in a CFST model using various grid sizes. (**a**) Case 3 (d = 1 cm); (**b**) Case 6 (d = 2 cm); (**c**) Case 7 (d = 0.5 cm).

**Figure 10 sensors-24-02503-f010:**
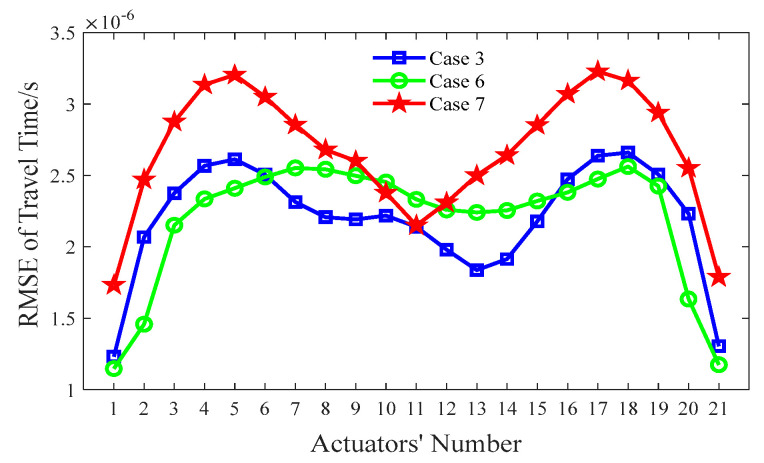
RMSE of travel time with various *d*s.

**Figure 11 sensors-24-02503-f011:**
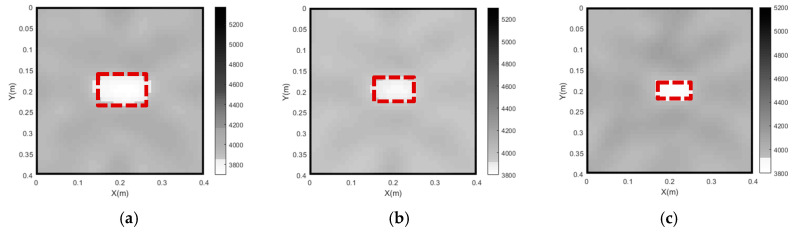
The tomography results of the single concrete core void defect of CFST models with various void defect sizes/ratios. (**a**) 6% void ratio; (**b**) 3.75% void ratio; (**c**) 2% void ratio.

**Figure 12 sensors-24-02503-f012:**
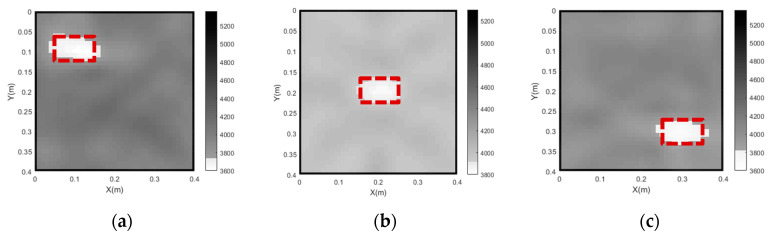
The tomography results of a single concrete core void defect located at different positions. (**a**) defect in the left upper corner; (**b**) defect in the center; (**c**) defect in the right lower corner.

**Figure 13 sensors-24-02503-f013:**
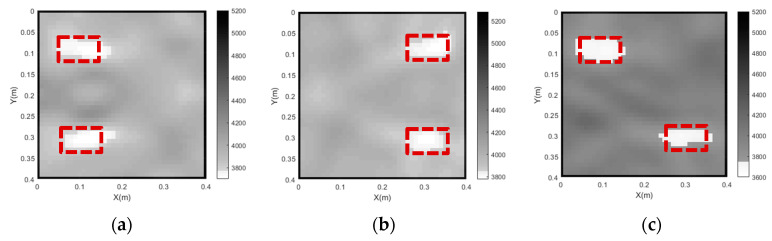
The tomography results of two concrete core void defects located at different positions. (**a**) Near the actuator side; (**b**) near the sensor side; (**c**) along the diagonal line.

**Figure 14 sensors-24-02503-f014:**
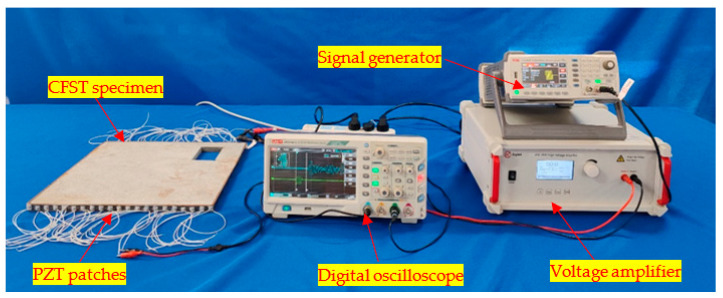
The test setup.

**Figure 15 sensors-24-02503-f015:**
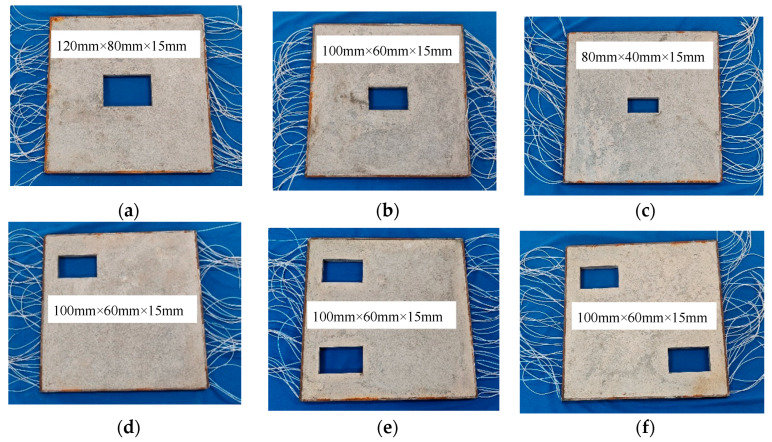
Six CFST specimens with various void defects. (**a**) CFST-S1; (**b**) CFST-S2; (**c**) CFST-S3; (**d**) CFST-S4; (**e**) CFST-S5; (**f**) CFST-S6.

**Figure 16 sensors-24-02503-f016:**
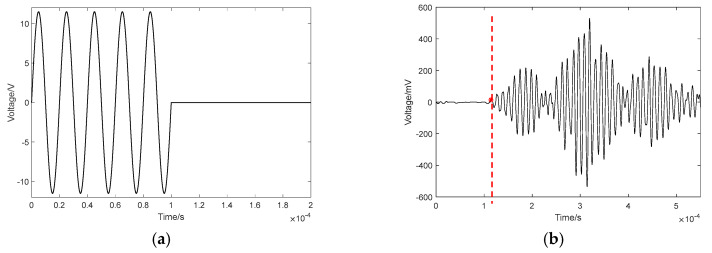
The sinusoidal impulse modulation excitation and stress wave response signal. (**a**) Sinusoidal impulse modulation excitation; (**b**) the stress wave response of the PZT sensor.

**Figure 17 sensors-24-02503-f017:**
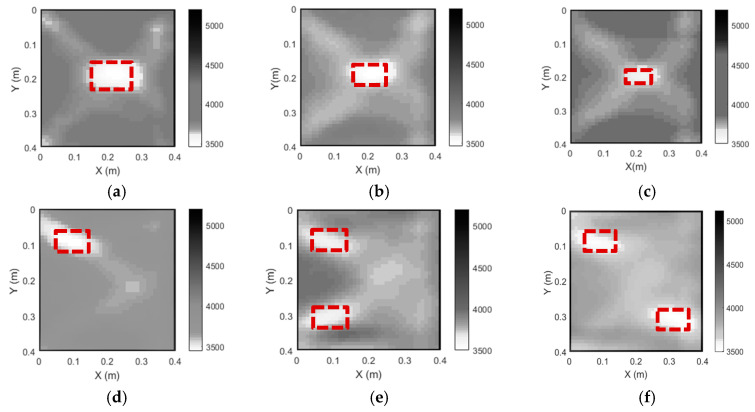
The tomography results of six CFST specimens with various void defects. (**a**) CFST-S1; (**b**) CFST-S2; (**c**) CFST-S3; (**d**) CFST-S4; (**e**) CFST-S5; (**f**) CFST-S6.

**Table 1 sensors-24-02503-t001:** Material properties of the PZT-CFST coupling model.

Number	Material	Elastic Modulus (GPa)	Poisson Ratio	Density (kg/m³)
1	Steel	206	0.30	7850
2	Concrete	30.0	0.17	2400
3	PZT5A	---	---	7500

**Table 2 sensors-24-02503-t002:** The seven cases and their selected different parameters.

Cases	$W_{m}$	λ	*d*
Case 1	Unit matrix operator	5	1 cm
Case 2	first-order difference operator	5	1 cm
Case 3	second-order difference operator	5	1 cm
Case 4	second-order difference operator	2	1 cm
Case 5	second-order difference operator	10	1 cm
Case 6	second-order difference operator	5	2 cm
Case 7	second-order difference operator	5	0.5 cm

## Data Availability

Data are contained within the article.

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
