# Peer review of "A Concrete Core Void Imaging Approach and Parameter Analysis of Concrete-Filled Steel Tube Members Using Travel Time Tomography: Multi-Physics Simulations and Experimental Studies"

_sensors, 2024, doi:10.3390/s24082503_

Round 1

Reviewer 1 Report (New Reviewer)

Comments and Suggestions for Authors

The submitted Article with the Manuscript ID: sensors-2887442 and the Title: “A concrete core void imaging approach for concrete-filled steel tube members and parameter analysis with travel time tomography: Multi-physics simulation and experimental studies” introduces a concrete core void imaging approach for concrete-filled steel tube (CFST) members using a curved ray theory-based travel time tomography (TTT) method. The study tries to visualize and quantify the sizes and locations of void defects with acceptable accuracy through a combination of numerical simulations with finite element models and experimental validation on CFST specimens. Parameter analysis is conducted to optimize the imaging algorithm, highlighting the influence of the model weighting matrix, weighting factor, and grid size on the imaging. To conclude, the proposed technique aims to demonstrate its effectiveness in detecting internal defects in CFST structures, thus offering valuable insights for enhancing the damage investigation of civil engineering applications. The material expounded in the manuscript is comprehensive, showcasing a well-organized and eloquently articulated framework. Nevertheless, specific segments necessitate careful examination and refinement to align with requisite scientific standards. The following remarks and recommendations are provided for the authors' consideration and guidance:

1. Although the introduction provides essential foundational information, the scope of the literature review needs to be expanded. To enhance the research objectives, consider incorporating the following relevant topics:

(a) Since the study uses a Non-destructive evaluation technique to determine the material's elemental composition, a description of the latest developments in the experimental investigation of cementitious elements, focusing on damage investigation and monitoring with different Non-destructive Evaluation techniques, could be helpful

(b) The utilization of non-destructive techniques can also detect damage and voids in structural elements. Worldwide, methods for detecting voids on reinforced concrete elements have already been established and are highly accepted in the construction industry. Meanwhile, considering the scattering of the fracture energy in the microstructure is important for understanding the crack path during the distribution of damage in the fracture surface.

Undertaking a systematic literature review in accordance with the recommendations could prove advantageous in this context.

The subsequent articles, identified for their relevance, could serve as illustrative examples to support the aforementioned issues:

- "Acoustic monitoring for the evaluation of concrete structures and materials," in Acoustic Emission and Related Non-Destructive Evaluation Techniques in the Fracture Mechanics of Concrete (Second Edition), 2021.

- "Electromechanical properties of multi-reinforced self-sensing cement-based mortar with MWCNTs, CFs, and PPs," Construction and Building Materials, 2023.

2. All abbreviations should be described at the beginning of usage.

3. All the variables and the equations presented should be described and appropriately named. A notation list presented in a new table could be helpful.

4. A more detailed theoretical explanation is advised for the main results.

5. What are the limitations of the proposed approach? ​

6. In Figure 12, the third dimension should also be included, even if plate-shaped.

7. Lacks a clear introduction section that provides background information on CFST members and the significance of concrete core void imaging. ​

Author Response

Reviewer 2 Report (New Reviewer)

Comments and Suggestions for Authors

Several comments should be considered.

1. The related literature should be added for some equations.

2. How to validate the accuracy of the method.

3. Several figures should be improved to be more clear, e.g. Fig. 13a.

4. The English writing should further improved.

Comments on the Quality of English Language

Some sentences should be further improved. 

Author Response

Reviewer 3 Report (New Reviewer)

Comments and Suggestions for Authors

Title: Visualizing and Quantifying Defects in Concrete Filled Steel Tube (CFST) Members: A Critical Analysis

Decision: Minor Revision

The study focuses on visually representing and quantifying defects within Concrete Filled Steel Tube (CFST) members, particularly concrete core voids. While the research offers valuable results, several weaknesses in the current document need to be strengthened to ensure its scholarly value equals the importance of the publication.

Chapter 1

1. A more in-depth critical analysis or comparison of existing methods in the literature review can enhance the reader's depth of understanding. The authors may add more state-of-art articles in engineering application for the integrity of the manuscript (3D vision technologies for a self-developed structural external crack damage recognition robot; Automation in Construction. An experimental investigation and machine learning-based prediction for seismic performance of steel tubular column filled with recycled aggregate concrete. Reviews on Advanced Materials Science).

2. While the introductory section establishes a solid foundation for the study by introducing the topic, outlining research gaps, and stating objectives, improving clarity and conciseness can further enhance the paper's impact and readability.

Chapter 2

3. The article lacks real cases or application examples to verify the effectiveness and practicality of the numerical simulation process.

4. The mathematical derivations in the article may be too lengthy and require simplification or additional explanation and clarification.

Chapter 3

5. While the article describes the extraction method for the first wave peak arrival time, it lacks a detailed discussion of the method's validity and accuracy, which could be appropriately included.

Chapter 4

6. The article analyzes the effects of parameter selection but lacks an in-depth discussion of the influence mechanisms. It is recommended that the authors further explore the influence mechanisms of different parameters on imaging results and provide a more comprehensive analysis.

7. The authors should provide more quantitative analysis results and quantitative comparisons of imaging effects under different parameters.

Chapter 5

8. The authors should discuss the scope of application, limitations, and potential directions for improving the methodology to enhance its usefulness and reliability.

Chapter 6

9. This section adequately describes the experimental setup, including specimen preparation, sensor arrangement, and excitation signal generation, aiding in understanding the experimental procedure.

10. The analysis of possible errors and uncertainties in the experimental process and their impact on the imaging results can be beneficial.

Chapter 7

11. The paper should suggest research directions and possible extensions to guide the reader in considering the future development of the research.

12. This can include improvements to current research limitations, exploration of other materials, or applications in different structural types, among others.

Round 2

Reviewer 1 Report (New Reviewer)

Comments and Suggestions for Authors

The revised Article with the Manuscript ID: “sensors-2887442-v2” and the title: “A concrete core void imaging approach for concrete-filled steel tube members and parameter analysis with travel time tomography: Multi-physics simulation and experimental studies” has been improved extensively. The efforts performed by the Authors to consider all the recommendations and to respond to all the criticisms of the previous review are greatly appreciated. It is outstanding that all of the them have been considered sincerely and have been responded successfully. Hence, it is suggested to be accepted for publication as it is.

This manuscript is a resubmission of an earlier submission. The following is a list of the peer review reports and author responses from that submission.

Round 1

Reviewer 1 Report

Comments and Suggestions for Authors

1.The article studies the imaging method of voids in rectangular cross-section steel tube concrete. Is it suitable for circular cross-section steel tube concrete?

2.The shrinkage and non axial stress of concrete in steel tube concrete can lead to voids between the concrete and the steel tube. In practical engineering applications, the expansion of core concrete under stress will offset voids. How to image this working condition?

3.The concrete voids in the finite element model are in a very ideal state. How to identify non-uniformly distributed voids?

Comments on the Quality of English Language

Few spelling and grammar errors need to be corrected.

Reviewer 2 Report

Comments and Suggestions for Authors

Concrete-filled steel tube (CFST) members are extensively employed in civil engineering due to their exceptional mechanical properties. However, inherent flaws like concrete core voids and interface debonding can undermine their load-bearing capacity and stiffness, impacting the safety and utility of CFST structures. Addressing the critical need to detect internal defects in CFST construction, this study introduces travel time tomography (TTT) to quantitatively identify and visualize CFST member dimensions and positions. Additionally, a parameter analysis investigates the relationship between TTT imaging quality and factors such as inversion parameters, defect sizes, and positions. Numerical examples validate the effectiveness and accuracy of the TTT algorithm, highlighting its efficiency in identifying concrete core void defects and the significant influence of inversion parameters, including model weighting matrix and inversion grid size, on imaging outcomes. This study well presents optimal parameters to advance the prospective TTT approach for CFST members.

Reviewer's Comments:

In response to the reviewer's comments:

1. Reason for Choosing Rectangular CFST Shape:

The decision to employ a rectangular shape for the CFST members, deviating from the conventional circular shape, needs to be clarified. Elaborate on the rationale for this choice, its alignment with the study's objectives, and its potential implications on the research outcomes.

2. References for Material Property Values in Table 1:

Please ensure that proper references are provided for the material property values listed in Table 1. Each value should be substantiated by citing reliable sources that support the chosen material properties.

3. Explanation of Base Software/Framework for TTT:

Provide a comprehensive explanation of the underlying software/framework used for implementing the travel time tomography (TTT) method. Whether it is MATLAB, Python, or another library, detail the specific tools and libraries utilized to ensure the reproducibility of the TTT algorithm.

4. Examples of Equipment for TTT Measurements:

Present specific examples of the equipment used for conducting TTT measurements. Describe the types of sensors or devices employed, their configurations, and operational principles. This information is crucial for comprehending the measurement setup and ensuring the study's reliability.

Incorporate these suggestions to enhance the quality and clarity of your manuscript.
